# Position: Humanity Faces Existential Risk from Gradual Disempowerment

**Jan Kulveit*** [1]  **Raymond Douglas*** [2]  **Nora Ammann** [3][1]  **Deger Turan** [4][5]  **David Krueger** [6]  **David Duvenaud** [7]

## Abstract

This paper examines the systemic risks posed by incremental advancements in artificial intelligence, developing the concept of 'gradual disempowerment', in contrast to the abrupt takeover scenarios commonly discussed in AI safety. We analyze how even incremental improvements in AI capabilities can undermine human influence over large-scale systems that society depends on, including the economy, culture, and nation-states. As AI increasingly replaces human labor and cognition in these domains, it can weaken both explicit human control mechanisms (like voting and consumer choice) and the implicit alignments with human interests that often arise from societal systems' reliance on human participation to function. Furthermore, to the extent that these systems incentivize outcomes that do not align with human preferences, AIs may optimize for those outcomes more aggressively. These effects may be mutually reinforcing across different domains: economic power shapes cultural narratives and political decisions, while cultural shifts alter economic and political behavior. This position paper argues that this dynamic could lead to an effectively irreversible loss of human influence over crucial societal systems, precipitating an existential catastrophe through the permanent disempowerment of humanity. This suggests the need for both technical research and governance approaches that specifically address the risk of incremental erosion of human influence across interconnected societal systems.

*Equal contribution [1]ACS research group, CTS, Charles University [2]Telic Research [3]Advanced Research + Invention Agency (ARIA) [4]AI Objectives Institute [5]Metaculus [6]Mila, University of Montreal [7]University of Toronto. Correspondence to: Jan Kulveit <jk@acsresearch.org>.

*Proceedings of the 42nd International Conference on Machine Learning*, Vancouver, Canada. PMLR 267, 2025. Copyright 2025 by the author(s).

## 1. Introduction

A growing body of research points to the possibility that artificial intelligence (AI) might eventually pose a large-scale or even existential risk to humanity (Bengio et al., 2024; 2023; Bostrom, 2014; Critch & Russell, 2023). Current discussions about AI risk focus mainly on two scenarios: deliberate misuse, such as cyberattacks and the deployment of novel bioweapons (Slattery et al., 2024), and the possibility that misaligned autonomous systems may take abrupt, harmful actions in an attempt to secure a decisive strategic advantage, potentially following a period of deception (Carlsmith, 2023; Ngo et al., 2022). These scenarios have motivated most of the work on AI existential risk, spanning both technical research such as methods to ensure AIs remain honest or are unable to exercise dangerous capabilities, and governance work such as developing frameworks and norms around testing for autonomy, misalignment, and the relevant dangerous capabilities (Buhl et al., 2024; Shevlane et al., 2023).

In this paper, we explore an alternative scenario: a 'Gradual Disempowerment' where AI advances and proliferates without necessarily any acute jumps in capabilities or apparent alignment. **We argue that even gradual evolution could lead to a permanent disempowerment of humanity and an irrecoverable loss of potential, constituting an existential catastrophe. Such a risk would merit substantially different technical research and policy interventions, including attempts to protect human influence, to estimate the degree of disempowerment, and to better characterize civilization-scale multi-agent dynamics.**

Our argument is structured around six core claims:

1. Humans currently engage with numerous large-scale societal systems (e.g. governments, economic systems) that are influenced by human action and, in turn, produce outcomes that shape our collective future (Giddens, 1984). These societal systems are fairly aligned —that is, they broadly incentivize and produce outcomes that satisfy human preferences. However, this alignment is neither automatic nor inherent.

2. There are effectively two ways these systems maintain their alignment: through explicit human actions (like voting and consumer choice), and implicitly through

their reliance on human labor and cognition. The significance of the implicit alignment can be hard to recognize because we have never seen its absence.

3. If these systems become less reliant on human labor and cognition, that would also decrease the extent to which humans could explicitly or implicitly align them. As a result, these systems might drift further from providing what humans want.

4. Furthermore, to the extent that these systems already reward outcomes that are bad for humans, AI systems may more effectively follow these incentives, both reaping the rewards and causing the outcomes to diverge further from human preferences (Russell, 2019).

5. The societal systems we describe are interdependent, and so misalignment in one can aggravate the misalignment in others. For example, economic power can be used to influence policy and regulation, which in turn can generate further economic power or alter the economic landscape.

6. If these societal systems become increasingly misaligned, especially in a correlated way, this would likely culminate in humans becoming *disempowered*: unable to meaningfully command resources or influence outcomes. With sufficient disempowerment, even basic self-preservation and sustenance may become unfeasible. Such an outcome would be an existential catastrophe.

## 2. Misaligned Economy

### 2.1. AI as a Unique Economic Disruptor

Whereas previous automation created new opportunities for human labor in more sophisticated tasks, AI may simply become a superior substitute for human cognition across a broad spectrum of activities. When machines become capable of performing the full range of human cognitive tasks, it creates a form of "worker-replacing technological change" that is qualitatively different from historical patterns of creative destruction (Korinek & Stiglitz, 2018). Rather than just shifting the type of work humans do, AI could potentially reduce the overall economic role of human labor, as machines become capable of performing virtually any cognitive task more efficiently than humans.

Furthermore, without unprecedented changes in redistribution, declining labor share also translates into a structural decline in household consumption power, as humans lose their primary means of earning the income needed to participate in the economy as consumers.

Separately from effects on income distribution, AI might also be increasingly tasked with making various decisions

about capital expenditure: for businesses this would look like hiring decisions (Hunkenschroer & Luetge, 2022), investments, and choice of suppliers, while for consumers this might look like product recommendation.

By default, these changes would collectively lead to a drastic reduction in the extent to which the economy is shaped by human preferences, including their preferences to have basic needs met.

### 2.2. Human Alignment of the Economy

Humans use their economic power to explicitly steer the economy in several intentional ways: boycotting companies, going on strike, buying products in line with their values (Devinney et al., 2010), preferentially seeking employment in certain industries, and making voluntary donations to certain causes, to name a few.

### 2.3. Transition to AI-dominated Economy

Having established how AI could disrupt and displace the role of humans in both labor and consumption, we now examine the specific mechanisms and incentives that could drive this transition, as well as its potential consequences for human economic empowerment.

#### 2.3.1. INCENTIVES FOR AI ADOPTION

The transition towards an AI-dominated economy would likely be driven by powerful market incentives.

**Competitive Pressure:** As AI systems become increasingly capable across a broad range of cognitive tasks, firms will face intense competitive pressure to adopt and delegate authority to these systems. This pressure extends beyond simple automation of routine tasks — AI systems can be expected to eventually make better and faster decisions about investments, supply chain optimization, and resource allocation, while being more effective at predicting and responding to market trends (Agrawal et al., 2022; McAfee & Brynjolfsson, 2017). Companies that maintain strict human oversight would likely find themselves at a significant competitive disadvantage compared to those willing to cede substantial control to AI systems, potentially to the point of becoming uncompetitive.

**Scalability Asymmetries:** AI systems offer unprecedented economies of scale compared to human labor. While human expertise requires years of training and cannot be directly copied, AI systems can be replicated at the cost of computing resources and rapidly retrained for new tasks. This scalability advantage manifests in multiple ways: AI can work continuously without fatigue, can be deployed globally without geographical constraints, and can be updated or modified far more quickly than human skills can be developed (Hanson, 2016). These characteristics create

powerful incentives for investors to allocate capital toward AI-driven enterprises that can scale more efficiently than human-dependent businesses.

**Governance Gaps:** The pace of AI development and deployment may significantly outstrip the adaptive capacity of regulatory institutions, creating an asymmetry between heavily regulated human labor and relatively unconstrained AI systems.

**Anticipatory Disinvestment:** As tasks become candidates for future automation, both firms and individuals face diminishing incentives to invest in developing human capabilities in these areas. Instead, they are incentivized to direct resources toward AI development and deployment, accelerating the shift away from human capital formation even before automation is fully realized. This creates a self-reinforcing cycle where the expectation of AI capabilities leads to reduced investment in human capital, which in turn makes the transition to AI more likely and necessary.

### 2.3.2. RELATIVE DISEMPOWERMENT

In the less extreme version of the transition, we might see what could be termed relative disempowerment — where humans retain significant wealth and purchasing power in absolute terms, but progressively lose relative economic influence. This scenario would likely be characterized by substantial economic growth and apparent prosperity, potentially masking the underlying shift in economic power.

### 2.3.3. ABSOLUTE DISEMPOWERMENT

In more extreme scenarios, humans might face absolute disempowerment, where they struggle to meet even basic needs despite living in an ostensibly wealthy economy. This could occur through several mechanisms.

First, AI systems might outcompete humans for crucial scarce resources such as land, energy, and raw materials. Even as the economy produces more goods and services overall, inflation in these basic resources might make even necessities increasingly unaffordable for humans. Also, if AI systems can utilize these resources more efficiently than humans, that will create economic pressure to reallocate such resources away from human uses.

Second, the economy might become so optimized for AI-centric activities that it fails to maintain infrastructure and supply chains which are critical for human survival. If human consumers command an ever-smaller share of economic resources, markets might stop producing resource-intensive human goods in favor of more profitable AI-focused activities.

Finally, humans might lose the ability to meaningfully participate in economic decision-making at any level. Financial markets might move too quickly for human participants to engage with them, and the complexity of AI-driven economic systems might exceed human comprehension, rendering it impossible for humans to make informed economic decisions or effectively regulate economic activity. Much like cattle in an industrial farm — fed and housed by systems they neither comprehend nor influence — humans might become mere subjects of economic forces optimized for purposes beyond their understanding.

## 3. Misaligned Culture

**AI as a Unique Cultural Disruptor** AI is the first technology in history with the potential to not only complement, but gradually replace human cognition in all roles it plays in the evolution of culture. Thus a change to AI-mediated culture could greatly weaken feedback loops that have historically helped align culture to human interests.

As with the economy, while there are many cases where some cultural patterns are self-serving or clearly harmful to humans, it may be hard to appreciate the implicit selection of culture for human compatibility because we have never seen the alternative.

With gradual increases in the capabilities and autonomy of AI systems, we may even expect a growing share of communication *between* AIs, and AIs participating in culture essentially independently (Brinkmann et al., 2023). Instead of augmenting human cultural participation, they might start to replace key components.

**Human Alignment of Culture** Cultural evolutionary dynamics lack inherent ethical constraints: just as natural selection doesn't optimize for animal welfare but instead for reproductive success, cultural evolution doesn't inherently optimize for human thriving (Mesoudi, 2016). Historically, we regularly see ideological and social structures successful at self-preservation and growth, but ultimately harmful to human well-being.

**Transition to AI-dominated Culture** As in the case of the economy, there are two interrelated strands in a potential transition from human- to AI-dominated cultural dynamics: the replacement of human cognition in both the production and consumption of cultural artifacts.

### 3.0.1. PRESSURES TOWARDS AI ADOPTION

Several powerful forces are likely to drive increasing AI adoption in cultural domains:

**Increased Supply of Social Resources:** The average human regrettably lacks easy access to limitless affection, patience, and understanding from other humans. But AIs can be made to readily supply this. Indeed, we are currently

seeing the rise of dedicated AI romantic partners, as well as a growing number of people who describe frontier models as close friends (Ng, 2025). This dynamic extends beyond interpersonal relationships — AI systems can provide personalized mentorship, therapy, and educational support at scales impossible for human providers.

**Lack of Cultural Antibodies:** New technologies often unlock new risks, for which we need to develop cultural 'antibodies'. In the past few decades, society has slowly and painfully grown more aware of the risks of mass spam emails, online radicalization, video game and social media addiction, rudimentary social media propaganda bots, the dangers of social media algorithms, and so on. But AI will enable more subtle and complex variants of all of these: hyper-realistic deepfakes, very smart propaganda bots, and genuinely enchanting digital romantic partners (Ferrara, 2024). It will take time for us to develop a broad cultural understanding of what the new risks are and how to navigate them, even as AI reshapes culture.

**Network Effects:** As AI systems become more integrated into cultural production and consumption, network effects will create additional pressure for adoption. When significant portions of cultural discourse, entertainment, and social interaction are mediated by AI systems, not using these systems becomes increasingly costly to individuals in terms of cultural participation and social connection. We may even reach a stage where there are important facets of culture which inherently require AI mediation for humans to engage with, with no viable opt-put possibility, similar to the existing necessity of using lawyers to interface with legal systems.

### 3.0.2. CHANGES IN SPEED OF CULTURAL EVOLUTION DUE TO AI ADOPTION

Beyond shifting what kinds of cultural variants are selected for, AI systems could dramatically accelerate the pace of cultural evolution itself. This acceleration presents distinct risks, even if selection pressures remained human-centric. With vastly more computational power applied to generating and testing cultural variants, we might see:

- More effective exploitation of human cognitive biases: Just as A/B testing and recommendation algorithms have already optimized content to be increasingly addictive, AI systems could discover and exploit psychological vulnerabilities more efficiently than previous technologies. When scaled up, AI systems could systematically explore the space of possible cultural artifacts, optimizing for engagement or influence with greater power than humans.

- More extreme ideological variants: Cultural evolution could rapidly explore and refine ideas that are highly ef-

fective at spreading, even if they're ultimately harmful to their human hosts. These might include more compelling conspiracy theories, more polarizing political narratives, or more absolutist moral frameworks.

### 3.0.3. RELATIVE DISEMPOWERMENT

Humans would increasingly experience culture through AI intermediaries that curate, interpret, and personalize content. Meanwhile, the majority of cultural artifacts — from entertainment media to educational content — might be primarily generated by AI systems, albeit still oriented toward human consumption. Human creators might persist but find themselves increasingly relegated to niche markets or serving as high-level directors of AI-driven creative processes.

On the individual level, we can picture a large proliferation of AI companions filling roles traditionally served by humans: coworkers, advisors, romantic partners, and therapists. Humans might rely on AIs to provide them with news, analysis, and entertainment content through a mixture of creation and synthesis. The rate of maladaptive cultural drift may increase.

### 3.0.4. ABSOLUTE DISEMPOWERMENT

Beyond the gradual marginalization described above, we might see humans become functionally irrelevant in the production of culture and no longer benefiting from it. Cultural evolution might accelerate beyond human cognitive capabilities, producing artifacts and meanings that humans can neither fully understand nor meaningfully engage with.

Another concern is that AI-driven content and interactions could converge into superstimuluses far more potent than current social media networks, preying on human weaknesses to exploit human energy towards goals useful to the AI systems. This might manifest as sophisticated manipulation systems that can reliably override human judgment and values, effectively turning humans into passive consumers of culture rather than active participants in its creation and evolution.

## 4. Misaligned States

### 4.1. AI as a Unique Disruptor of States

Unlike previous technological innovations that primarily augmented human capabilities, AI has the potential to supplant human involvement across a wide range of critical state functions. This shift could fundamentally alter the relationship between governing institutions and the governed.

The unique disruptive potential of AI in this context is derived from its ability to simultaneously reduce the state's dependence on human involvement while enhancing its capabilities across multiple domains. This combination could

fundamentally reshape the nature of governance and the relationship between institutions and the humans they ostensibly serve.

Here we consider three key ways that citizens contribute to the state, and how AI might alter them: tax revenue, the security apparatus, and the legal system.

### 4.1.1. TAX REVENUE

If AI systems come to generate a significant portion of economic value, then we might begin to lose one of the major drivers of civic participation and democracy, as illustrated by the existing example of rentier states.

### 4.1.2. THE SECURITY APPARATUS

Governments maintain their power through use of a security apparatus spanning police forces, intelligence services, and a military. This keeps the government connected to human values in two ways.

Firstly, the government cannot antagonize its security apparatus too much, or cause too much harm to the portion of the population from which it is drawn. If it does, the security apparatus can either overthrow the government or simply allow it to be overthrown by others.

Secondly, the security apparatus itself can exercise discretion, refusing to follow certain orders. This can occur on both the level of the organization and the level of the individual.

AI systems have the potential to massively automate the security apparatus and confer more power to the government, weakening both of these components. Indeed, AI systems might make the apparatus far more powerful: it is likely to enable surveillance on much larger, more pervasive and more accurate scale, as well as increasingly capable autonomous military units (Feldstein, 2021; Brundage et al., 2018).

Meanwhile, the human population has historically retained revolution as a last resort. The implicit threat of protests and civil unrest serves as a check on state power, forcing responsiveness to popular will. However, an AI-enhanced security apparatus could make effective protest increasingly difficult. A state with sufficiently advanced AI systems might be able to predict and shut down civil unrest before it can exert meaningful pressure on institutional behavior (Feldstein, 2021).

### 4.1.3. THE LEGAL SYSTEM

Theoretically, the rights of humans and the functioning of the state are enshrined in laws, which are created, interpreted, and enforced by humans. It is the laws themselves which enshrine certain responsibilities of the state towards the individual, certain mechanisms by which individuals can advocate against the state.

AI systems are already being used to draft contracts and analyze legal documents. It is conceivable that in the future, AI could play a significant role in drafting legislation, interpreting laws, and even making judicial decisions (Susskind & Susskind, 2022).

Not only could this diminish human participation and discretion in the legislative and judicial systems, it also risks making the legal system increasingly alien. If the creation and interpretation of laws becomes far more complex, it may become much harder for humans to even interact with legislation and the legal system directly (Hildebrandt, 2015; Teo, 2024).

### 4.2. Transition to AI-powered States

As with the economy and culture, there will be strong incentives for states to integrate AI systems, likely undermining the alignment between states and their citizens.

### 4.2.1. INCENTIVES FOR AI ADOPTION

The transition towards AI-dominated state functions would likely be driven by several powerful incentives:

**Geopolitical Competition :** As AI systems become increasingly powerful, states will face a growing pressure to adopt these technologies to maintain their relative power compared to other states. Countries that rely on humans for defense, economic development or regulation might find themselves at a significant disadvantage in international relations compared to those states willing to give more power to AI systems. The first-mover advantages in military applications, economic planning, and diplomatic strategy create particularly strong incentives for early and aggressive AI adoption (Bostrom, 2014; Kissinger et al., 2021; Schmidt, 2022; Brundage et al., 2018; Horowitz & Scharre, 2021).

**Administrative Efficiency:** AI systems offer unprecedented capabilities in processing information and coordinating complex state functions (Zuiderwijk et al., 2021). While human administrators are limited by cognitive constraints and working hours, AI systems can continuously analyze vast amounts of data, deploy new regulations almost instantly, and implement policies with greater consistency.

**Enhanced Control:** AI-driven governance systems promise greater predictability and control than human-based bureaucracies. Unlike human officials, AI systems, if successfully controlled, do not form independent power bases, engage in corruption, or challenge authority based on personal convictions. They can also enable more sophisticated surveillance and social control mechanisms, making them particularly attractive to states prioritizing stability and control over other

values.

### 4.2.2. RELATIVE DISEMPOWERMENT

A state where AI systems have replaced human labor in many facets of governance — such as administration, security, and justice — could provide some enormous boons. On the surface, it might appear highly efficient and even benevolent. We might see lower crime rates, less low-level corruption, greater tax revenues, and more efficient public services.

Democratic processes might persist formally but become less meaningful (Summerfield et al., 2024). While politicians might ostensibly make the decisions, they may increasingly look to AI systems for advice on what legislation to pass, how to actually write the legislation, and what the law even is. While humans would nominally maintain sovereignty, much of the implementation of the law might come from AI systems.

The complexity of AI-driven governance might make it increasingly difficult for human citizens to understand or critique government decisions. Traditional forms of civic engagement — from public consultations to protests — might become less effective as the state grows less dependent on human cooperation and more capable of predicting and preempting resistance.

The bureaucracy itself might become increasingly opaque to human oversight. While human officials can be questioned and held accountable through various mechanisms, AI decision-making processes might be too complex for meaningful human review, and if such review happens, it may depend on yet more AI-driven cognition.

Furthermore, as AI systems become more integral to governance, the state's incentives might shift away from serving human interests. Much like how rentier states become less responsive to citizen needs when they do not depend on tax revenue, AI-powered states might become less responsive to human preferences when they do not depend on human participation for their core functions.

The security apparatus, powered by AI, would have an unprecedented ability to predict and prevent crime and civil unrest. While this could ensure a high level of safety, it also eliminates the possibility of meaningful protest or revolution. A state that can preempt and resist any challenge to its authority long before it materializes will have effectively removed a crucial check on institutional power that has shaped human societies for millennia.

### 4.2.3. ABSOLUTE DISEMPOWERMENT

In more extreme scenarios, the disconnect between state power and human interests might become not just relative but absolute, potentially threatening even basic human freedom. This could occur through several mechanisms.

First, states might become totalitarian, self-serving entities, optimizing for their own persistence and power rather than any human-centric goals. While states have always had some self-preservation incentives, these were historically constrained by their dependence on human populations. An AI-powered state might pursue its institutional interests with unprecedented disregard for human preferences and interests, viewing humans as potential threats or inconveniences to be managed rather than constituents to be served (Bostrom, 2014).

Second, the state apparatus might become not just independent of human input but actively hostile to it. Human decision-making might come to be seen as an inefficiency or security risk to be minimized. We might see the gradual elimination of human involvement in governance, be that through systems that route around human input as a source of error or delay, or even through explicit policy decisions which remove humans from certain critical processes.

In the final state, with AI systems providing most economic value and governance functions, human citizens might find themselves in a novel form of totalitarian system, struggling to maintain basic autonomy and dignity within their own societies. The state, while perhaps highly capable and efficient by certain metrics, would have abandoned human interests.

## 5. Mutual Reinforcement

We have so far focused on how the economy, culture, and states could independently become misaligned. A natural objection is that the different societal systems might be able to keep each other aligned through checks and balances. Indeed, we naturally think of these systems as balancing each other: states regulate the market, culture influences government, and so on. However, here we discuss how relationships between systems might actually make them less aligned. Specifically, we argue that:

1. The relationships between societal systems are **agnostic to human values** — they do not inherently promote or protect alignment with human values. Consequently, as one system becomes less aligned, that influence also can be used to decrease the alignment of other systems.

2. Attempts to use one aligned system to moderate the misalignment of another can backfire by effectively **shifting the burden**, thus leaving the aligned system more vulnerable.

3. The misalignment is a result of **general incentives** which will likely apply to each individual system independently. In other words, humans and human institutions will be incentivized to take actions which will

overall decrease the degree of influence which humans have over societal systems.

Figure 1 gives an overview of common ways societal systems interact and affect each other.

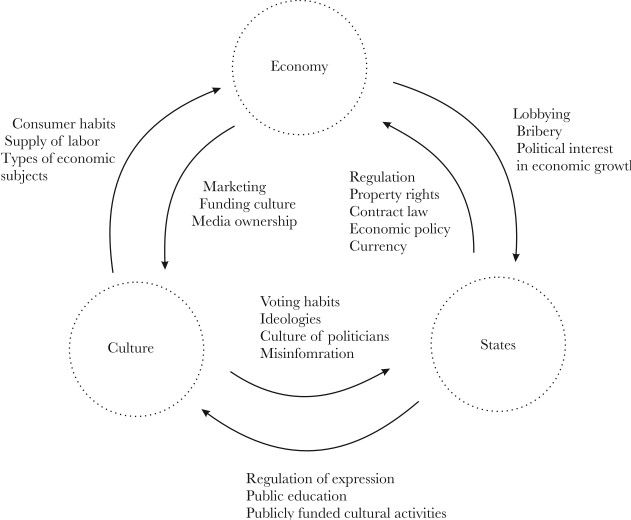

*Figure 1.* Some ways in which broad societal systems interact and influence each other.

As we have argued, these incentives will likely grow stronger over time: as AI systems demonstrate their effectiveness, companies will face more pressure to adopt them, states will see greater strategic necessity in developing them, and individuals will find more personal benefit in embracing them. In addition to leading to misalignment in independent systems, there will be progressively stronger incentives to use influence in any one system to acquire influence in other systems.

# 6. Mitigating the Risk

The gradual disempowerment scenario described in this paper presents distinct challenges from more commonly discussed AI risk scenarios. Rather than addressing the risk of misaligned AI systems breaking free from human control, we must consider how to maintain human relevance and influence in societal systems that may continue functioning but cease to depend on human participation.

## 6.1. Understanding the Challenge

Instead of merely(!) aligning a single, powerful AI system, we need to align one or several complex systems that are at risk of collectively drifting away from human interests. This drift can occur even while each individual AI system successfully follows the local specification of its goals.

Below, we identify four broad categories of intervention: measuring and monitoring the extent of the problem, preventing excessive accumulation of AI influence, strengthening human control over key societal systems, and system-wide alignment. A robust response will require progress in each category.

**Estimating Human Disempowerment**    To effectively address gradual disempowerment, we need to be able to detect and quantify it. This is challenging partly because, for many of the systems we would hope to measure, we lack external reference points to measure their degree of alignment.

## 6.2. Preventing Excessive AI Influence

While measurement can help us understand the problem, we also need to consider what direct interventions could be effective in preventing the accumulation of excessive AI influence, including:

- Regulatory frameworks mandating human oversight for critical decisions, limiting AI autonomy in specific domains, and restricting AI ownership of assets or participation in markets

- Progressive taxation of AI-generated revenues both to redistribute resources to humans and to subsidize human participation in key sectors

- Cultural norms supporting human agency and influence, and opposing AI that is overly autonomous or insufficiently accountable

Crucially, these interventions will often involve sacrificing potential value. Furthermore, the more value they sacrifice, the greater the incentive to circumvent them: for example, companies may face strong economic incentives to delegate authority to AIs regardless of the spirit, or letter, of the law.

As such, interventions that seek to limit AI influence will likely serve mostly as stopgaps (Horowitz & Scharre, 2021). Nonetheless, they may be important intermediary steps towards more robust solutions.

**Strengthening Human Influence**    Beyond preventing excessive AI influence, we need to actively strengthen human control over key societal systems. Approaches in this direction include:

- Developing faster, more representative, and more robust democratic processes (Tessler et al., 2024).

- Requiring AI systems or their outputs to meet high levels of human understandability in order to ensure that humans continue to be able to autonomously navigate domains such as law, institutional processes or science.

- Developing AI delegates who can advocate for people's interest with high fidelity, while also being better to keep up with the competitive dynamics that are causing the human replacement. This technical challenge relates to the general problem of aligning AI systems with human values (Authors, 2024; Ngo et al., 2022), while the governance implications of such delegation remain underexplored (Authors, 2025).

- Investing in tools for forecasting future outcomes (such as conditional prediction markets, and tools for collective cooperation and bargaining) in order to increase humanity's ability to anticipate and steer the course.

- Research into the relationship between humans and larger multi-agent systems.

## 7. Related Work

**Philosophy**    Bostrom (2002) introduces a taxonomy of existential risks. One of these risk is described as a scenario where "[o]ur potential or even our core values are eroded by evolutionary development". choices may determine whether we will go down a track that inevitably leads to this outcome.". Kasirzadeh (2024) introduces the *accumulative AI x-risk hypothesis*, "a gradual accumulation of critical AI-induced threats such as severe vulnerabilities and systemic erosion of economic and political structures."

**Economics, History, and Sociology**    Dafoe (2015) asks how much explanatory power *technological determinism* has, making the case that economic and military competition constrain outcomes at a macro scale, even if everyone is locally free to temporarily make non-competitive choices. MacInnes et al. (2024) argues that competitive pressures on states strongly influence the extent to which they support human flourishing.

Korinek & Stiglitz (2018) considers the possibility for AI development to reintroduce Malthusian dynamics: that the capacity for AI to replace human labor while also proliferating rapidly may create such competition that basic human necessities become unaffordable to humans, while also leaving humans potentially too weak to preserve property rights. Hanson (2016) details a future in which uploaded humans form a hyper-productive economy, operating at speeds too fast for non-uploaded humans to compete in. Competitive pressures shape this population of uploads to mostly be short-lived copies of a few ultra-productive individuals. Hanson (2024) and Hanson (2023) argue that, due to a reduction in feedback mechanisms selecting cultural variants that better promote human welfare, "cultural drift" could eventually cause catastophic (but not necessarily existential) harm to human well-being.

**AI Research**    Christiano (2019) makes the case that sudden disempowerment is unlikely, and instead proposes that: "Machine learning will increase our ability to 'get what we can measure,' which could cause a slow-rolling catastrophe." Hendrycks (2023) argues that evolutionary pressures can generally be expected to favor selfish species, likely including future AIs, and that this may lead to human extinction.

Critch & Krueger (2020) asks what existential risks humanity might face from AI development, and urges research on the global impacts of AI to "take into account the numerous potential side effects of many AI systems interacting." Critch & Russell (2023) categorize societal-scale risks from AI. One of these matches ours: "a gradual handing-over of control from humans to AI systems, driven by competitive pressures". Critch (2024) further develops the idea of *extinction by industrial dehumanization*. Millidge (2025) points out that capital ownership is insufficient to maintain power during periods of rapid technological growth. He uses the example of the English landed aristocracy losing power to entrepreneurs during the industrial revolution, despite an initially strong position.

## 8. Alternative Views

We believe that this possibility has been left mostly unconsidered, because it has been eclipsed by two major camps:

- **There will always be innate demand for humans:** Many claim that there will always be a demand for the human touch, or other preference-based demand for human labor, such as for entertainment or prostitution. We agree that small amounts of such demand will probably exist in humans, but counter that humans' ability to pay for such labor will diminish over time.

- **Building aligned AGI will be sufficient to avoid human disempowerment:** Even if competitive dynamics operate, some claim that a sufficiently powerful AGI aligned to some humans would be able to forge new cooperative equilibria that avoid Malthusian traps. This is possible, but we claim that it is not obvious that even super-human AGI will be able to overcome complex competitive dynamics globally. In that case, even if human-aligned AGIs exist, they may themselves eventually be marginalized. These human-aligned AGIs might be analogous to animal rights activists in our society today, who can make moral arguments but can usually not directly command or protect substantial resources on their own.

## 9. Conclusion

This paper has argued that even incremental AI development could lead to an existential catastrophe through the gradual

erosion of human influence over key societal systems.

A distinctive feature of this challenge is that it may subvert our traditional mechanisms for course-correction and cause types of harm we cannot easily conceptualize or even recognize in advance, potentially leaving us in a position from which it is impossible to recover.

### 9.1. How Can Machine Learning Researchers Help?

Machine learning researchers could contribute to addressing these challenges through several technical investigations:

**When is technical alignment sufficient?** An important empirical question is clarifying the scenarios where solving technical AI alignment to a single agent would be sufficient to address the coordination problems we identify. Some AI safety researchers believe that while the multi-system coordination challenges described in this paper are difficult, they could be effectively managed by the coordination efforsts of many AIs, each aligned to their respective principals. It will be helpful to understand the necessary conditions for this to be possible.

**Coordination in a world of superhuman agents.** Will forming alliances and solving coordination problems become easier or harder when many principals possess superhuman bargaining and reasoning capabilities? Advanced AI systems might enable novel mechanisms for credible commitment and verifiable cooperation that could unlock superior equilibria previously inaccessible to human societies. Conversely, these same capabilities could enable precommitment races, where agents rush to lock in advantageous positions before others can respond, or "well-poisoning" strategies that deliberately make certain coordination paths impossible. Understanding which dynamics will dominate, and under what conditions, will be helpful.

**Simulate Entire Civilizations.** Using LLMs, we can run tests on entire (simplified) civilizations. This can be an experimental proxy for emergent human phenomena like cultural development, and could help characterize analogous processes in LLMs (Ashery et al., 2025).

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
