# OpenReview forum: "Position: Humanity Faces Existential Risk from Gradual Disempowerment"
_ICML.cc/2025/Position_Paper_Track — ICML 2025 Position Paper Track poster_

### Official Review · Reviewer_j1dN · 2025-02-21

**Significance:** 4
**Argument Clarity:** 4
**Rating:** 5
**Confidence:** 4

**Questions:**

I would like to recommend some additional directly relevant references that I think could help expand topics that already exist in this work. Let me know if you agree.

(1) Cooperative human-AI interactions and democracy:

Tessler, Michael Henry, et al. "AI can help humans find common ground in democratic deliberation." Science 386.6719 (2024): eadq2852.

Summerfield, Christopher, et al. "How will advanced AI systems impact democracy?." arXiv preprint arXiv:2409.06729 (2024).

(2) Inability of AI (LLMs at least) to capture human diversity:

Atari, Mohammad, Mona J. Xue, Peter S. Park, Damián E. Blasi, and Joseph Henrich. “Which Humans?”, (2023) https://psyarxiv.com/5b26t
See Fig 3 especially

Wang, A., Morgenstern, J. & Dickerson, J.P. Large language models that replace human participants can harmfully misportray and flatten identity groups. Nat Mach Intell (2025). https://doi.org/10.1038/s42256-025-00986-z

(3) Social, academic and geopolitical effects of the militarization of AI:

Simmons-Edler, Riley, et al. "Position: AI-Powered Autonomous Weapons Risk Geopolitical Instability and Threaten AI Research." (2024) Forty-first International Conference on Machine Learning.

Shah, Raj M., and Christopher Kirchhoff. Unit X: How the Pentagon and Silicon Valley Are Transforming the Future of War. Simon and Schuster, 2024.

Horowitz, Michael C., and Paul Scharre. AI and International Stability. CNAS, 2021.

Scharre, Paul. Army of None: Autonomous Weapons and the Future of War. Vol. 52. WW Norton & Company, 2018.

Chahal, H., Fedasiuk, R., and Flynn, C. Messier than oil: Assessing data advantage in military AI. CSET Issue Brief, 2020.

(4) AI replacing human counselors and romantic partners

Ng, Kelly. 'DeepSeek moved me to tears': How young Chinese find therapy in AI. (2025) BBC.

(5) Dangers of over-trusting / over-incorporating AI even when AI cannot perform well, e.g. certain dangers your paper proposes don't actually require high performing AI if humans over-trust lower performing AI.

Holbrook, Colin, et al. "Overtrust in AI Recommendations About Whether or Not to Kill: Evidence from Two Human-Robot Interaction Studies." Scientific reports 14.1 (2024): 19751.

Messeri, Lisa, and M. J. Crockett. "Artificial intelligence and illusions of understanding in scientific research." Nature 627.8002 (2024): 49-58.

**Discussion Potential:**

4

**Paper Summary:**

This paper examines the broader and possibly very detrimental effects that increasingly AI-centered economies, and social & civic spaces will have on human culture. Importantly, they discuss possible cultural and political dangers that can occur from increasing incorporation of AI into our governments and workplaces even if the superhuman AGI/ASI is not achieved soon and even if the worst cases of misalignment are avoided. Warning signs are already seen in the present day of many of the social changes the paper discusses, and they combine and integrate them in a comprehensive way with literature support. This position paper argues that "AI safety" is currently overly narrow as a focus, and that computer scientists need to interact with a broader sphere of policy makers and social scientists to design more comprehensive AI safety roadmaps that address more immediate possible social effects.

**Position:**

Yes

**Position In Title:**

Yes

**Related Work:**

3

**Strengths And Weaknesses:**

This topic will certainly encourage a lot of discussion as written, and will hopefully help AI safety efforts and experts on the computer science side better understand the AI safety concerns from the view of policy makers, the public, and social scientists more deeply to enhance collaboration. Alternative views and self-debates are provided.

There is a risk current AI safety is perhaps overly focused on (1) avoiding long-term speculative sci-fi level, superhuman AI existential risks from 'god-like' AI, and (2) more grounded but still narrow short-term metrics of safety evaluations for corporate LLMs. While scattered blogs and papers discuss various serious secondary social effects such as automation replacing human labor (usually shallowly analyzed), I have not yet read a work that combines the different cultural/social AI topics within one position-oriented paper for a computer science audience in such a comprehensive and still deep way, within the confines of a conference proceedings paper length anyways. I think framing AI safety in the broader and updated framework the authors have put forward will be useful to have at ICML.

In the end section discussing solutions and new broader metrics, I noticed the citation amount got sparse and often focused on older (though still appropriate) works. The authors and general computer science audience may not be aware of more recent directly relevant work in this topic area going on at the boundary of social sciences and computer sciences and should reexamine the literature. I have provided some examples in the "Questions" section below to aid this effort.

Last, this paper captures well cultural and national level regulatory and norms problems, and touches upon follow-up technical work that should be done for those interested in the position. However, for the ICML computer science audience, I strong recommend adding a more specific section about what computer scientists in academia and industry specifically should do in this area as well. Are certain topics or types of collaborations especially important to give more attention and priority to?

**Support:**

4

---

> ### Author Rebuttal · Authors · 2025-04-01
>
> Thank you for the detailed and helpful feedback, especially the proposed citations!
>
> We plan to add all of the proposed citations to a long-form version of this paper.  Due to space limitations, we plan to add the following citations from your list:
>  - Tessler, Michael Henry, et al. "AI can help humans find common ground in democratic deliberation." Science 386.6719 (2024).
>  - Summerfield, Christopher, et al. "How will advanced AI systems impact democracy?." arXiv preprint arXiv:2409.06729 (2024).
>  - Horowitz, Michael C., and Paul Scharre. AI and International Stability. CNAS, 2021.
>  - Ng, Kelly. 'DeepSeek moved me to tears': How young Chinese find therapy in AI. (2025) BBC.
>
> We’re not yet sure about adding "Artificial intelligence and illusions of understanding in scientific research.", since we expect to also face the even harder problem of deserved reliance on AI.  We’re also not sure about the citations on the difficulties current AI has with representing the diversity of human values, since we don’t think the technical difficulty of doing so will be the main bottleneck in the long run.
>
> As for how ICML computer scientists could help address these problems:  Great question!  Here are two somewhat technical questions that they might be able to help with:
> 1) In what scenarios could solving technical alignment be sufficient?  Some AI safety researchers seem to believe that the sorts of coordination problems pointed to in our paper are difficult, but could and would be addressed by sufficiently-well-aligned AIs acting on our behalf.  Clarifying the conditions necessary for that to be true seems like an important empirical question.
> 2) Will solving coordination problems and forming alliances become easier or harder in a world where many principals have superhuman bargaining and reasoning abilities, or special ways to prove honest commitment?  It could be that this creates many superior equilibria, or that it allows precommitment races and well-poisoning at scale.
>
> We will add these last suggestions to the paper.
>
> Besides these, ICMLers could help with:
>  - Doing large-scale analysis of what AI is being used for and what impact it's having.
>  - Doing analysis of e.g. how culture spreads in AI, what benchmarks capture our concerns, etc.
>  - Figuring out how we can use AI differentially to empower humans / stabilise the world, and actually implementing it.
>  - Doing research on “ecosystem alignment”, a sort of whole-civilization game theory.

---

> > ### Comment · Reviewer_j1dN · 2025-04-02
> >
> > Thank you for your response. I find the author rebuttal sufficient to address my concerns.

---

### Official Review · Reviewer_RDaZ · 2025-03-14

**Significance:** 4
**Argument Clarity:** 3
**Rating:** 4
**Confidence:** 4

**Questions:**

What makes AI-driven economic displacement fundamentally different from previous automation waves, which societies have historically adapted to?

How do you account for potential regulatory, legal, or social responses that could prevent disempowerment?

You assume that AI systems will increasingly optimize for objectives misaligned with human well-being. What specific mechanisms could counteract this drift within AI development?

**Discussion Potential:**

4

**Paper Summary:**

The paper argues that AI poses an existential risk not through sudden, catastrophic failure modes but via gradual disempowerment, where human influence over critical societal systems -- economic, cultural, and political -- is systematically removed. The authors suggest that AI's increasing role in decision-making, economic allocation, and governance could lead to human irrelevance, as AI systems optimize for incentives that do not necessarily align with human well-being. The paper outlines mechanisms by which this disempowerment could unfold, including economic displacement, AI-driven cultural shifts, and governance automation, and highlights the self-reinforcing nature of these processes across different societal domains. Unlike traditional AI risk narratives focused on misalignment or sudden power shifts, this perspective emphasizes a gradual but irreversible trajectory towards human exclusion from meaningful decision-making and resource control.

**Position:**

Yes

**Position In Title:**

Yes

**Related Work:**

3

**Strengths And Weaknesses:**

The strength of the paper is its reframing of existential AI risk beyond catastrophic failure scenarios. The focus on gradual systemic changes offers a fresh and new perspective that is often overlooked in AI safety discussions. The argument is well-structured, and the breakdown of economic, cultural, and political disempowerment mechanisms provides a comprehensive theoretical framework for considering long-term AI risks. Additionally, the paper successfully highlights the interdependence of societal systems, demonstrating how AI-driven misalignment in one domain (e.g., economic automation) can accelerate disempowerment in others (e.g., political influence, cultural relevance).

One weakness is the unclear distinction between economic and existential risks. The paper shows loss of economic agency with complete human irrelevance, assuming that AI dominance in production will necessarily exclude humans from influence. However, historical automation trends suggest that societies adjust to labor displacement through new economic and political structures, even in cases of extreme technological transformation. The paper does not convincingly argue why AI-driven economic shifts would be uniquely irreversible compared to previous industrial revolutions.

**Support:**

3

---

> ### Author Rebuttal · Authors · 2025-04-01
>
> Thank you for the detailed feedback.  We fear you might have missed one of our main arguments, but fortunately you got to the heart of the matter in your review:  “What makes AI-driven economic displacement fundamentally different from previous automation waves, which societies have historically adapted to?“
>
> The answer, we claim, is that AGI could act as a replacement, rather than a complement, for almost all human labor.  This would put humans in an analogous situation to horses after the invention of superior sources of mechanical power.
>
> As for your question about predictable responses that might reverse such a trend:  We spend much of the paper arguing that specific countermeasures such as AI bans or re-skilling would either merely delay disempowerment, or be routed around by competitive pressures.  Section 5 in particular addresses this point.  However, this does not mean that such approaches aren’t worthwhile to try!
>
> As for your question about more effective potential responses:  Section 6, “Mitigating the Risk”, contains our list of proposed research and policy responses to this problem.  Specifically, we outline ways to more explicitly align incentives between the state and humans.

---

### Official Review · Reviewer_njn1 · 2025-03-14

**Significance:** 4
**Argument Clarity:** 4
**Rating:** 5
**Confidence:** 4

**Questions:**

How does this model of gradual disempowerment differ fundamentally from existing discussions on AI autonomy and bureaucratic control?
Can you provide historical examples or empirical data to support the claim that technological displacement has led to irreversible societal disempowerment?

**Discussion Potential:**

4

**Paper Summary:**

The paper argues that---in addition to the well-known disruptive risks of advanced general AI systems discussed in the AI safety literature, such as large-scale misuse by bad human actors or abrupt loss of control due to misaligned AI systems---there might also be a substantial existential risk related to slowly enfolding, gradual, but irreversible disempowerment of humans due to several complex, interrelated, and self-reinforcing processes. It describes in detail scenarios in which humans might loose control of economic, legal, political, social, and cultural systems, and might loose access to key resources such as labor income, social interaction, meaning, or scarce resources. It finally calls for focusing on better understanding the challenge and strengthening human influence.

**Position:**

Yes

**Position In Title:**

Yes

**Related Work:**

3

**Strengths And Weaknesses:**

*Strengths*

The paper reads extremely well and does not require specialized background knowledge.

It has a very clear structure, first describing potential causal processes that would lead to gradual disempowerment in individual global systems, then describing relevant reinforcing interactions between these systems, shortly discussing opposing views, and finally hinting at possible mitigations (which are not the focus of the paper).

Even though each individual described process is highly uncertain, they make an excellent point that these processes are highly plausible, would have existential consequences for humans, and so their possibility must be taken seriously.

Strong Framing of AI Risks: The concept of gradual disempowerment highlights how AI-driven shifts in economic, political, and cultural systems could subtly disempower humans without requiring a direct takeover.

Interdisciplinary Approach: The paper successfully integrates AI safety concerns with socio-economic and governance perspectives, broadening the discourse beyond technical alignment.

Compelling Systemic Analysis: The discussion of feedback loops between AI-driven economic incentives, cultural influence, and governance structures is well-articulated and thought-provoking.

*Weaknesses*

Not Entirely Novel: While the terminology is new, similar concerns have been explored in discussions on AI's impact on human autonomy, governance, and economic displacement in the AI safety literature. Prior work on AI-enhanced decision-making in bureaucracies, warfare, and politics already touches on some of these risks.

Lack of Empirical Support: The argument is mostly theoretical, with limited quantitative modeling to support claims about AI’s long-term societal impact. This would however probably be too much to ask for. However, some sections could have profited from a few more references.

**Support:**

4

---

> ### Author Rebuttal · Authors · 2025-04-01
>
> Thank you for the feedback.  We agree with your characterization of the paper’s weaknesses.  We believe we’ve covered the most closely related work, but are all ears for suggestions for any we’ve missed, and plan to add several references suggested by Reviewer j1dN.
>
> As for your questions:  Our model is different from existing discussions on AI autonomy and bureaucratic control in that we emphasize that existing characterizations of state and institution’s responses to improvements in autonomy understate the risk.  This is because existing experience has all been gathered in a context where humans were necessary for operations.
>
> As for historical examples, it would be almost impossible to have witnessed this happening to humans already, although one could argue that it has happened to some groups of extinct proto-humans.  We can point to specific groups becoming unrecoverably politically weak, partly through competitive pressures forcing external alliances - perhaps the decline of the Tokugawa Shogunate or Native Americans are analogous examples.  A less severe but related phenomenon today is that lots of tiktok/instagram users would prefer that everyone stop, but do not feel they can unilaterally stop: https://bfi.uchicago.edu/insight/research-summary/when-product-markets-become-collective-traps-the-case-of-social-media/

---

### Official Review · Reviewer_PS8g · 2025-03-14

**Significance:** 3
**Argument Clarity:** 3
**Rating:** 4
**Confidence:** 4

**Questions:**

Question 1: Could the authors highlight any smaller-scale or near-term examples of “gradual disempowerment” that help illustrate the core mechanism more concretely?

Question 2: The idea of “AI delegates” that truly represent human interests is intriguing. Could the authors cite some relevant work on technical or governance safeguards needed to ensure these delegates remain genuinely human-aligned?

**Discussion Potential:**

4

**Paper Summary:**

This paper argues that humanity could lose meaningful control over critical societal systems—such as the economy, culture, and states—not through a sudden AI “takeover,” but through slow, incremental adoption of increasingly capable AI systems. According to the authors, as AI automates more human cognitive tasks and decision-making, humans will gradually lose both explicit and implicit means of steering these systems. The paper contends that this drift could lead to large-scale misalignment and eventual “disempowerment” of humanity. The authors suggest that the community should broaden technical and policy measures to address not just abrupt failure modes (e.g., a single misaligned superintelligence) but also the more insidious risk that many smaller AI systems, operating in concert, erode human influence over time.

**Position:**

Yes

**Position In Title:**

Yes

**Related Work:**

3

**Strengths And Weaknesses:**

Strength: The paper states a distinct position, highlighting a potential existential risk rooted in slow societal shifts rather than abrupt, singular AI-driven disasters. It underscores the need to develop institutional checks and monitoring protocols, a research direction often overshadowed by purely technical “alignment solutions.”

Weakness 1: The paper's claim that AI systems could exacerbate misalignments is based on mutually reinforcing interactions across different domains (i.e., economy, culture, and states). While Figure 1 provides a useful overview of the mechanisms linking these entities, it only presents them as keywords (e.g., voting habits, ideologies) without further elaboration. A more detailed discussion of these interactions in Section 5 would strengthen the argument. Additionally, these relationships have been extensively studied in prior social science research, yet relevant citations are missing. Including references to these works would help ground the discussion in existing literature.

Weakness 2: The terms "relative" and "absolute" disempowerments are interesting, but the paper does not provide clear definitions for these concepts.

**Support:**

3

---

> ### Author Rebuttal · Authors · 2025-04-01
>
> Thank you for the detailed feedback.  You seem to clearly understand our main argument.
>
> Re: Weakness 1: We agree that the details of interaction mechanisms need to be fleshed out to strengthen our argument, and to shed more light on potential remedies.  We regard this as future work.
>
> As for citations, we did cite Dafoe (2015) on technological determinism, MacInnes et al. (2024) on competitive pressures on states, and Korinek & Stiglitz (2018) from economics.  But we agree that citing more related work would strengthen the paper, and are all ears for suggestions.  We spent 1/16 of the paper on related work, but perhaps should have spent more space or been briefer.  We plan to add several references suggested by Reviewer j1dN.
>
> Re: Weakness 2:  Good point.  We plan to add the following sentence:  “We begin each section by describing mechanisms that could cause partial, or relative disempowerment, then conclude with mechanisms that could cause total, or absolute, disempowerment.”
>
> Re: Question 1: We’re happy to point to some examples — one case of ongoing disempowerment is the weakening of legacy media in the face of content recommendation algorithms. Historically, newspapers had a lot of resources and therefore a lot of room to hire journalists and exercise editorial judgment; since the rise of social media and targeted advertising they’ve been compelled to become a lot more polarised and sensational. Of course the picture here is complex, but one story you can tell is that a machine alternative is creating competitive pressures which make it harder for human discretion and judgment to influence our culture.
>
> Another near-term risk is the increasing use of automated drone weaponry: it seems plausible that we’ll see a race for adoption, and for automation higher and higher up the chain, which will in turn remove a certain amount of human discretion from warfare. There are already some controversies over automatic target recommendation in drone strikes [1].
>
> We also give some examples of a slightly different nature in our response to reviewer njn1, which may be of interest.
>
> Re: Question 2: We agree that AI delegates represent an interesting opportunity. The technical problem probably reduces to the general challenge of aligning an AI to the values of a human, which is well-discussed in the literature - for instance in [2] and [3]. The governance question is less explored, although a paper was published after our submission on how AI might affect liberal governments, which touches on some of the challenges of delegation, accountability, and the balance of power [4]. There’s an interesting puzzle we allude to in the paper about the balance between being aligned to any given human, and being collectively aligned to humanity. We’ll gladly add these references to the paper, and also emphasise that a lot of work remains to be done.
>
> [1] https://www.972mag.com/lavender-ai-israeli-army-gaza/
> [2] https://arxiv.org/abs/2404.09932
> [3] https://arxiv.org/abs/2209.00626
> [4] https://arxiv.org/pdf/2503.05710

---

> > ### Comment · Reviewer_PS8g · 2025-04-02
> >
> > Thanks for the clarifications and proposed additions. I agree that incorporating the citations suggested by reviewer j1dN would help make the pathways in Fig. 1 more concrete and better anchored in existing research. One minor typo: "Misinfomration" in Fig. 1.

---

### Decision · Program_Chairs · 2025-04-27

**Decision:**

Accept (poster)

**Comment:**

We have updated the decision on this paper based on the reviews, meta-reviews, and lively discussion that points to the value of this timely and controversial topic. In addition to other revisions, the authors should carefully address the limitations noted in the area chair's meta-review, to make the scenarios more concrete, and to think carefully about actionable recommendations.

---
Original meta-review:

This paper argues that rather than an immediate catastrophe due to an AGI the humanity will gradually become disempowered. This position is supported by a discussion a variety of societal scenarios. The compilation of these plausible scenarios could be a useful resource for anticipating and mitigating some of the risks. Indeed reviewers have liked this aspect of the work. At the same time, while most scenarios are certainly plausible, they are not described sufficiently concretely to allow for a concrete evaluation of risk (let alone useful discussion of mitigation strategies). Let me mention that the basic premise of this paper that our community (and research more broadly) focuses on abrupt takeover by AGI is a strawman argument. In particular, there is plenty of work and discussion out there on the anticipated changes in the labor markets, economics, politics, and the gradual "irrelevance" of humans. In general, few (including myself) doubt that superhuman AI will bring about a dramatic change to human life with a variety of old and new risks. It is also well recognized that most of these changes are very hard to foresee and thus hard to mitigate now.  So overall, the main issue with this work is lack of new and specific insights about the interaction between AI and humans that are described at a level that lends itself for a technical discussion about potential mitigation. Essentially the paper discusses the future of extremely complex and unpredictable systems at the level of a main steam media essay.
It should also be noted that a more technical discussion of risks is entirely possible and there is plenty of technical work on the milder and more insidious risks from the use of AI (and algorithms more generally) For example the community around : https://facctconference.org/